# Comparison of a Vintage and a Recently Released Nematicide for the Control of Root-Knot Nematodes and Side Effects on Two Entomopathogenic Nematodes

**DOI:** 10.3390/plants10081491

**Published:** 2021-07-21

**Authors:** Ioannis O. Giannakou, Stefanos Kamaras

**Affiliations:** Laboratory of Agricultural Zoology and Entomology, Department of Science of Crop Production, Agricultural University of Athens, Iera Odos 75, 11855 Athens, Greece; stkamaras@gmail.com

**Keywords:** nematicidal activity, plant parasitic nematodes control, EPN, root-knot nematodes management

## Abstract

Root-knot nematodes can cause tremendous losses in vegetable crops. Farmers usually rely on synthetic nematicides to protect their crops. Recently, newly released nematicides are giving farmers an alternative in chemical control for nematodes. In the present study, the efficacy of vintage nematicide was compared to that of a relatively new nematicide, fluopyram. The latter was always more effective in substantially lower concentrations than oxamyl. Fluopyram paralyzed more than 80% J2s after 24 h immersion at the concentration of 0.25 μL L^−1^, while the percentage was increased close to 100% after immersion for 48 and 96 h. Similar levels of dead J2s were observed after immersion of J2s in oxamyl at concentrations higher than 8 μL L^−1^ (24 and 48 h) or 4 μL L^−1^ (96 h). An evident decrease of egg differentiation was observed when fluopyram concentration was increased to 8 μL L^−1^, while no significant decrease in egg differentiation was recorded at any concentration of oxamyl. Egg hatching was decreased at concentrations of fluopyram higher than 4 μL L^−1^, while no reduction was observed even when the concentration of oxamyl was increased to 64 μL L^−1^. The efficacy of fluopyram in soil was superior compared to that of oxamyl. For the first time, the systemic action of fluopyram is recorded in trials with tomato plants. On the other hand, compared to oxamyl, fluopyram seems to be more toxic to non-target organisms such as the entomopathogenic nematodes *Steinernema feltiae* and *Heterorhabditis bacteriophora*.

## 1. Introduction

Greenhouse cultivated vegetable crops are suffering from root-knot nematodes (RKN) [1,2]. RKN control has become increasingly important after the reduction of chemical nematicides use [3]. In Greece, application of nematicides remains the most common strategy for the control of RKN in greenhouses [4,5]. Although chemical nematicides provide superior management of nematodes, their use has been questioned in recent years because of increasing concern about environmental contamination [6], human health risks [7,8], high application costs [3], dependence of nematicide action on soil conditions, as well as increasing governmental regulation [9]. All these have created social and legislative pressure to remove many agricultural pesticides from the market [8,10], of which the most known examples are methyl bromide and 1,3-dichloropropene [11].

After the removal of effective fumigant nematicides, farmers struggle to fill the management gap continuously using the same non-fumigant nematicides. Oxamyl, a non-fumigant nematicide, is still in use, registered with different commercial names and different granulated and liquid formulations. Oxamyl, under different commercial names, is probably the most known nematicide all over the world. It is being used either in protected cultivating vegetables such as solanaceous and cucurbitaceous species or in open field cultivations. Oxamyl could be considered as the exception of the group of organophosphates and carbamates which have been withdrawn from the market in recent years. The fact that this chemical is one of the few nematicides still in the market, and because farmers use this active ingredient every year, could create biodegradation problems and efficacy decrease [12]. There are many examples in the literature reporting the rapid biodegradation of nematicides in general or specifically for oxamyl in farms in many countries [13,14].

This situation has enforced chemical scientists to search for new molecules to find new nematicides or to test already registered substances such as fungicides, insecticides, and acaricides against plant parasitic nematodes [15,16,17]. Most recently, fluopyram, which is a fungicide known for many years and controls botrytis and powdery mildew in vegetables, has been tested against plant parasitic nematodes [17]. It is released in the market since 2017 with various commercial names such as Vellum^®^ prime, Vellum^®^ one, ILevo, which are registered in many countries all over the world. It is a succinate dehydrogenase inhibitor (SDHI) fungicide belonging to the pyridinyl-ethyl-benzamide group. The main advantage of this compound is its ecotoxicological profile. It has low toxicity for rats (LD_50_ > 2000 mg kg^−1^) and bees (LD_50_ > 100 μg per bee). Another advantage of this nematicide is its low recommended dosage preventing an adverse effect to the agroenvironment. Fluopyram reduced root galling on lima bean, but a 22% reduction in emergence was recorded [18] which could be due to the damage of the cotyledons of emerging seedlings [19]. Second-stage juveniles of *M. javanica* exposed to doses of fluopyram higher than 1.0 mg/L were immobilized after 24 h (>85.9%) [20]. Although there are studies dealing with the systemic activity of oxamyl [21,22,23], no data is presented in the literature for fluopyram.

The objectives of the present study were: (1) to ascertain the effects of fluopyram and oxamyl in a series of concentrations in three different incubation periods against juveniles of *M. javanica*, (2) to study the effect of fluopyram and oxamyl in egg differentiation and hatching of *M. javanica*, (3) to study either the direct effect of fluopyram and oxamyl in dare soil infested with *M. javanica* or its systemic effect using tomato plants, (4) to ascertain the effect of fluopyram and oxamyl on two entomopathogenic nematodes.

## 2. Results

### 2.1. Nematode Motility Bioassays

Figure 1 presents the results of J2s of *M. javanica* in the motility bioassays using either fluopyram or oxamyl in a series of concentrations. The percentage of dead J2s increased by increasing the incubation time or the concentration. Constantly, fluopyram has shown 100% paralysis for all concentrations higher than 2 mg/L. The effect of fluopyram at the lowest concentration of 0.25 mg/L resulted in more than 80% dead juveniles after 24 h of incubation. The percentage of dead J2s was increased when the exposure time was increased to 48 and 96 h. On the other hand, similar efficacy in juveniles’ motility was recorded when oxamyl was used in higher concentrations. Specifically, oxamyl concentration had to increase to 8 mg/L to record a percentage of dead juveniles higher than 95%. After 96 h of incubation, only concentrations higher than 4 mg/L showed 100% dead juveniles. Based on the results, it is clear that fluopyram is very effective even at the lowest concentration tested.

### 2.2. Effect on Egg Differentiation

Egg differentiation was not inhibited by low concentration of fluopyram (Figure 2). The concentration had to increase to 8 mg/L for significant differences to be recorded comparing to the untreated control. A drastic decrease of egg differentiation was recorded when the concentration was increased to 32 and 64 mg/L. There was no inhibition on egg differentiation when undeveloped eggs were immersed in different concentrations of oxamyl. A low decrease, significantly different only to the untreated control but not to the other treatments, was recorded when the concentration of oxamyl was increased to 32 and 64 mg/L.

### 2.3. Effect of Oxamyl or Fluopyram on Hatching of Meloidogyne javanica Eggs

Significantly fewer juveniles were hatched from eggs treated with 4 mg/L of fluopyram or higher (Figure 3). No further significant difference was recorded when fluopyram concentration was increased from 32 to 64 mg/L. Oxamyl did not inhibit J2s hatching at all concentrations used.

### 2.4. Effect of Oxamyl or Fluopyram on Meloidogyne javanica Juveniles in Soil

There was a gradual decrease in the number of nematodes per gram of root as the concentration of fluopyram was increased to 1 mg/L (Figure 4). Any further increase of the concentration resulted in zero levels of nematodes in tomato roots. On the other hand, a completely different pattern was revealed when oxamyl was used. There were nematodes isolated from roots even when oxamyl was used at the highest (64 mg/L) concentration. Significant differences, compared to the control treatment, were recorded only when the concentration of oxamyl was increased to 4 mg/L, while a drastic decrease was indicated when the concentration was increased to 16 mg/L. The lowest number of females per gram of root was recorded in plants developed in soil treated with 64 mg/L of oxamyl.

### 2.5. Systemic Effect of Fluopyram

#### 2.5.1. Treated Tomato Plants in Infested Soil

A gradual decrease in the total number of nematodes inside the roots was recorded by increasing the concentration of fluopyram (Figure 5). The number of nematodes in the control treatment was significantly different compared to those in all concentrations of fluopyram. A drastic decrease in the number of nematodes was recorded when the concentration was increasing to 8 mg/L. The number of nematodes per root was constantly low as concentration was increasing from 8 to 64 mg/L. No significant differences were revealed between the control treatment and the concentrations of oxamyl up to 8 μL L^−1^. By increasing oxamyl concentration from 8 to 64 mg/L, there was a constant decrease of nematode numbers in tomato roots.

#### 2.5.2. Infected Tomato Plants Transplanted into Fluopyram- and Oxamyl-Treated Soil

No significant differences were recorded between plants transplanted into fluopyram- and oxamyl-treated soil at concentrations of 0.25–8 mg/L. However, significant differences between the two nematicides were recorded as the concentration was increased to 16, 32, and 64 mg/L (Figure 6).

### 2.6. Effect on Entomopathogenic Nematodes In Vitro

There was a clear toxic effect of fluopyram on third-stage instars of entomopathogenic nematodes as this was recorded in the motility tests. Instars of *Steinernema feltiae* were dead at a percentage of 64–69% when they were incubated for 24 h in concentrations from 1 to 32 mg/L (Table 1). A further increase in toxicity (79% dead instars) was recorded by increasing the concentration to 64 mg/L. The same pattern of results was recorded after incubating third-stage instars of *S. feltiae* for 48 h. However, an increase of dead instars was recorded by increasing the incubation time to 96 h. This increase was more evident at the lowest concentrations of fluopyram (0.25 and 0.5 mg/L). Fluopyram was also toxic to third-stage instars of *Heterorhabditis becteriophora*. There was an increase of dead instars as the concentration increased from 2 to 64 mg/L after 24 h of incubation. The toxic effect of fluopyram on *H. becteriophora* was 100% at the concentration of 8 mg/L (48 h of incubation) or 2 mg/L (96 h of incubation). Oxamyl was less toxic to third-stage instars of both entomopathogenic nematodes tested. Specifically, the highest percentage (12%) of dead instars of *S. feltiae* was recorded at the concentration of 64 mg/L after 24 h of incubation, which was increased to 17% at the same concentration after 48 h of incubation (Table 1). When nematodes were incubated for 96 h, there was a gradual increase in the number of dead third-stage instars of *S. feltiae* as the concentration increased. The highest percentage of dead instars (50%) was recorded at the concentration of 64 mg/L. Much lower toxicity was recorded when *H. becteriophora* was used. The highest toxicity (7%) was recorded at the concentration of 64 mg/L after 96 h of incubation. However, it seems that there was no clear effect of oxamyl to *H. becteriophora* third-stage instars irrespective of concentration and incubation time.

## 3. Discussion

In this study, we conducted a comparison between an old (oxamyl) and a recently released nematicide (fluopyram). Specifically, we investigated the concentration- and time-dependent effect of fluopyram and oxamyl against *Meloidogyne javanica*. We also investigated the possible negative impact of these nematicides against two entomopathogenic nematodes.

We showed that a concentration as low as 0.25 or 0.5 mg/L was effective in J2s paralysis after exposure to fluopyram for 24 h. Our results are in agreement with those reported from Faske and Hurd [17] showing a percentage of dead *M. incognita* juveniles which resulted in tomato root protection after exposure to fluopyram for 2 h. A similar level of immobilization was also reported by Oka and Saroya [20] at the concentration of 0.5 μL L^−1^ of fluopyram after 24-h exposure of *M. javanica* juveniles, although this was evident only in one of the two trials conducted. The percentage of dead J2s increased as the concentration of fluopyram increased. When the incubation time increased to 48 and 96 h, a complete kill of J2s (100% dead J2s) was recorded at the concentration of 2 mg/L (48 h of incubation) or at concentration of 0.25 mg/L (96 h of incubation). On the other hand, it is clear that a higher concentration of oxamyl was always required to gain the same percentage of kill effect at the same time, compared to fluopyram. This was expected, since the nematicides tested have different modes of action. Oxamyl is a carbamate acting as an inhibitor of acetyloholynesterase, whereas flyopyram is an SDHI inhibitor [23,24].

To the best of our knowledge, there are no published data in the literature for the effect of either fluopyram or oxamyl on egg differentiation. It was not affected when oxamyl was used at concentrations from 0.25 to 16 mg/L. A slight decrease of egg differentiation, compared to the untreated eggs, was recorded when the concentration was increased to 32 and 64 mg/L. Fluopyram ceased egg differentiation at a higher percentage as the concentration was increased to 8 and 16 mg/L (30 and 40% less differentiated eggs). This percentage was even higher at concentrations of 32 and 64 mg/L (70 and 80% less differentiated eggs). Depending on the gap between the uprooting of the previous crop and the transplanting of the next, eggs might be either developed (larval stages) or undifferentiated (cells multiplication stage).

Our results show that no inhibition on egg hatching was recorded after immersion of egg masses in all tested concentrations of oxamyl. In contrast, a hatching inhibition was evident as the concentration of fluopyram was increased from 2 to 64 mg/L. Three-day incubation of eggs at a concentration of 5 mg/L of fluopyram resulted in 96% hatching inhibition [25]. However, this hatching inhibition was reversible, even in a very high concentration of 100 mg L^−1^, since the hatching rate was similar to that observed in the untreated control after the removal of the eggs into clean water [25]. In our trials, eggs inside undisturbed egg masses were kept immersed in the chemical solution for a longer period (seven days), and then they were moved to clean water. The prolonged period of immersion time probably resulted in irreversible action of fluopyram in egg hatching. Additionally, the fluopyram solution seems to be able to enter the gelatinous matrix of egg mass and act on the eggs.

A drastic reduction of J2s survival was recorded as the concentration of fluopyram in soil was increased to 1 mg/L. Complete protection against RKN was recorded in soil treated with fluopyram at a concentration of 2 mg/L or higher. On the other hand, slight efficacy on J2s was recorded at concentrations of oxamyl up to 8 mg/L. Further oxamyl concentration increase resulted in J2s survival decrease that was inferior to that observed in fluopyram treatments. Except from preliminary experiments testing the action of chemical compounds against juveniles and eggs, in vitro experiments using soil are always necessary to conduct to clarify the efficacy of molecules in conditions very close to real.

Both nematicides tested have been reported to have acropetally systemic activity [22,25]. Although nematode control is related to the systemic activity of oxamyl, this is not reported for fluopyram. Our results show that good protection was achieved when tomato roots were treated with fluopyram and transplanted in soil infested with J2s of *M. javanica*. On the other hand, once J2s entered the roots, it was difficult to reduce nematode population when plants were transplanted into soil treated with either oxamyl or fluopyram.

It is clear from the results reported that fluopyram is very toxic to entomopathogenic nematodes. This could be undesirable in case that the farmer uses an EPN formulation to control soil insects. It has been reported that fluopyram has shown high toxicity to the nematode fauna of the soil [26]. It affects all nematode feeding groups soon after application, and this toxicity was extended for a prolonged time in the season [26]. However, apart from the negative impact on free-living nematodes, we assume that in many cases, EPNs share the same niches with plant parasitic nematodes. Therefore, care should be taken, especially in commercial farms where the presence of entomopathogenic nematodes is known.

Although fluopyram is very effective, even at very low concentrations killing second-stage juveniles, this is not the case as far as egg differentiation and egg hatching is concerned. However, the farmer could keep irrigating the soil for some days after the end of the previous crop to enhance the eggs into hatching and maximize the efficacy of the nematicide. Modern agriculture will rely on new molecules with innovative modes of action. Commonly, these new molecules are superior to the vintage ones, considering that they are active in lower doses, cost less, and show lower toxicity [25]. We have to bear in mind that higher doses, especially in agrochemicals applied in the soil, always lead to higher levels of residues in subsoil water. A relatively new nematicide such as that reported in the present study seems to be promising as far as efficacy against nematodes is concerned, but the side effects to non-target organisms such as EPN are not negligible and must be taken into consideration.

## 4. Materials and Methods

### 4.1. Meloidogyne javanica Cultures

A population of *Meloidogyne javanica* originally obtained from tomato roots in Crete, Greece was reared on tomato (*Solanum lycopersicum*) Mill. cv. Belladonna and maintained in a temperature-controlled glasshouse at 23–28 °C. Eggs were extracted with 1% sodium hypochlorite solution [27]. Second-stage juveniles (J2s) from infected tomato roots were allowed to hatch in a modified Baermann funnel. All J2s hatched in the first 3 days were discarded and thereafter, J2s collected after 24 h were used in the experiments.

### 4.2. Nematode Motility Bioassays

Solutions of fluopyram and oxamyl were tested for J2s motility at concentrations of 0.25, 0.5, 1, 2, 4, 8, 16, 32, and 64 mg/L. The chemicals were serially diluted in distilled water to produce test solutions of the abovementioned concentrations. Distilled water served as control. Approximately 50 J2s were used per treatment well in Cellstar^®^ 24-well plates which were exposed to fluopyram and oxamyl solutions. All treated and control plates were covered with a lid, wrapped with aluminum foil, and incubated at 26 ± 1 °C. Juveniles were observed with the aid of an inverted microscope (Zeiss, Germany) at 100× magnification after 12, 24, 48, and 96 h and were ranked as either alive or dead. Dead juveniles were certified by disturbing them with a needle. Those that remained immotile and had a straight shape were considered as dead. The experiment was conducted twice, while every treatment was replicated 5 times.

### 4.3. Effect on Egg Differentiation

*M. javanica* eggs were isolated from infected tomato roots using hypochlorite solution and following the procedure described by [27]. A clean solution of eggs was collected after sieving the egg suspension through 53 and 38 μm sieves. The quantification of egg suspension was done using an inverted microscope (100×), and eggs were used directly for the bioassays. Each well was filled with a 1 mL suspension containing approximately 50 eggs exposed to chemicals solution. All wells were incubated at 24 ± 1 °C. Chemical solutions (either fluopyram or oxamyl) were tested on development of eggs at concentrations of 0.25, 0.5, 1, 2, 4, 8, 16, 32, and 64 mg/L. Eggs in clean water served as the control. Those eggs with cell division (one, two, or more cells) were considered undifferentiated, while those with a fully developed juvenile were considered differentiated. All treated and control plates were covered with a lid to eliminate water evaporation. Each treatment was replicated five times, while the experiment was conducted twice.

### 4.4. Effect of Oxamyl or Fluopyram on Hatching of Meloidogyne javanica Eggs

Mature egg masses (dark yellow) from a population of *M. javanica* originally obtained from Crete and reared on tomato (*Solanum lycopersicum*) Mill. cv. Belladonna and maintained in a temperature-controlled glasshouse at 23–25 °C were handpicked and placed in small extracting trays made by 6 cm Petri dishes. Solutions of either fluopyram or oxamyl (0.25, 0.5, 1, 2, 4, 8, 16, 32, and 64 mg/L) were added to each extracting tray to cover the egg mass. Egg masses submerged in clean water served as the control. Extracting trays were maintained for 7 days in an incubator at 25 ± 1 °C, and then test solutions were removed by gently washing them with tap water. Each egg mass was returned to the same extracting tray and filled with clean water, covered to avoid loss of water, and placed in the incubator. Hatched J2s were collected and counted every 7 days, and all extracting trays were filled with clean water. The experiment was terminated after 30 days when no more J2s were collected from the extracting trays. The experiment was performed twice, and each treatment was replicated five times.

### 4.5. Effect of Oxamyl or Fluopyram on Meloidogyne javanica Juveniles in Soil

The efficacy of fluopyram and oxamyl against *M. javanica* was evaluated using tomato seedlings, cv. Belladonna. Seedlings at the four-leaf stage were grown in soil collected from a greenhouse (Table 1) at Gargaliani village, Peloponnese. The soil was autoclaved to kill any plant parasitic nematodes and also microflora which could be responsible for any biodegradation of the chemicals used. One-liter plastic bags were filled with 1 kg of soil, and water solutions of the nematicides were applied to meet the desired concentrations. Soil was thoroughly mixed and maintained for 2 days at 22 ± 1 °C. Five plastic pots were filled with 200 g of the treated soil, and 1 mL of suspension containing 500 J2s was used for inoculation. Pots under control treatment contained only soil with J2s. Pots were covered with aluminum foil to avoid water evaporation and maintained at 22 ± 1 °C for 24 h to allow J2s come in contact with the chemicals. Then, a seedling of tomato (cv. Belladonna) at the four-leaf stage was transplanted into the center of each pot. All plants were placed in a growth room at 25 ± 1 °C and were uprooted 28 days later, stems were removed, and roots were gently washed free of soil. The roots were stained using acid fuchsin as described in Byrd et al. [28]. All developing stages of nematodes inside the roots were counted under a stereoscopic microscope at 12.5× magnification [29]. All treatments were replicated five times, while the experiment was conducted twice.

### 4.6. Systemic Effect of Fluopyram

#### 4.6.1. Treated Tomato Plants in Infested Soil

Four-weeks-old tomato seedlings grown in plastic pots (6 cm deep and 3 cm in diameter) were drenched with solutions of fluopyram or oxamyl (0.25, 0.5, 1, 2, 4, 8, 16, 32, and 64 mg/L). The same procedure was repeated 4 days later. Pots under control treatment were drenched with water. Plants were maintained for another 3 days in a growth room at 25 ± 1 °C, and roots were carefully washed free of soil. Plants were transplanted into pots containing 200 g of soil inoculated with 500 J2s of *M. javanica* each. The moisture content was about 40% of the WHC to create ideal soil conditions for nematode movement. To avoid the transplanting stress of plants and keep the soil moist for as long as possible, aluminum foil was used to cover the soil surface around the plant stem. Plants were maintained in the growth room for 25 days and then uprooted, the roots were stained using acid fuchsin as described in [28], and all developmental stages of nematodes were counted as described previously.

#### 4.6.2. Infected Tomato Plants Transplanted into Fluopyram- and Oxamyl-Treated Soil

J2s were collected using the hypochlorite method as described previously. Four-weeks-old tomato plants in small plastic pots (6 cm deep and 3 cm in diameter) were inoculated with 500 J2s of *M. javanica* each. Infected plants were maintained for 3 days in a growth room at 25 ± 1 °C, and the roots were carefully washed free of soil. Then, infected plants were transplanted into pots containing soil treated with fluopyram. Pots with soil irrigated with water served as the control treatment. One kilogram of sandy soil collected from a greenhouse was divided into plastic bags after being autoclaved. Water solutions of fluopyram at concentrations of 0.25, 0.5, 1, 2, 4, 8, 16, 32, and 64 mg/L were used to treat 1 kg of soil in each bag. Moisture level was standardized to 40% of the WHC. Soil was thoroughly mixed and maintained at 22 ± 1 °C for 24 h. Five plastic pots were filled with the treated soil, and one seedling was transplanted into the center of each pot. Aluminum foil was used to cover the soil around the plant stem to avoid water evaporation. All plants were placed in a growth room at 25 ± 1 °C and uprooted 28 days later, and the roots were gently washed free of soil. The roots were stained, and all nematode stages were counted as described previously.

### 4.7. Effect on Entomopathogenic Nematodes

The procedure described previously (“Nematode motility bioassays”) was repeated, but instead of *M. javanica*, two species of entomopathogenic nematodes were used. Specifically, *Steinernema feltiae* and *Heterorhabditis becteriophora* were used as non-target organisms to monitor the action of fluopyram and oxamyl. Both species were supplied by Koppert B.V (Rodenrijs, The Netherlands). The chemicals were serially diluted in distilled water to produce test solutions (0.25, 0.5, 1, 2, 4, 8, 16, 32, and 64 mg/L). Distilled water served as control. Approximately 50 third-stage infective instars were used per treatment well in Cellstar^®^ 24-well plates. Larvae were observed with the aid of an inverted microscope (Zeiss, Germany) at 100× magnification after 24, 48, and 96 h and ranked either as alive or dead.

### 4.8. Statistical Analysis

Data were subjected to one-way analysis of variance (ANOVA) using the General Linear Model (GLM). Treatment means were compared using the LSD test. Statistical analysis in all cases was conducted using the SAS statistical package. Whenever appropriate, two experiments were combined and analyzed together, if no variation was revealed between the data.

## 5. Conclusions

In conclusion, our work showed that oxamyl and fluopyram can control *M. javanica*. However, fluopyram was more effective than oxamyl against J2s of *M. javanica* in substantially lower concentrations. Decrease of egg differentiation along with the inhibition of egg hatching were observed only with fluopyram. The efficacy of fluopyram in soil was superior compared to that of oxamyl. On the other hand, compared to oxamyl, fluopyram was more toxic to the entomopathogenic nematodes *S. feltiae* and *H. bacteriophora*.

## Figures and Tables

**Figure 1 plants-10-01491-f001:**
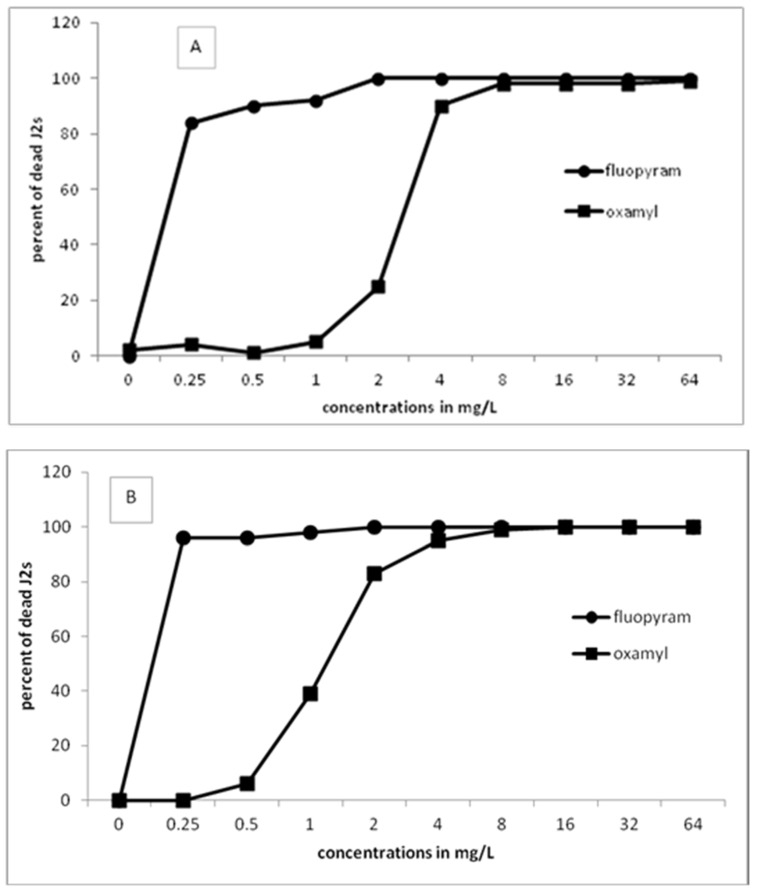
Effect of different concentrations of fluopyram or oxamyl on the motility of second-stage juveniles of *Meloidogyne javanica* after immersion of 50 J2s in test solutions for 24 (**A**), 48 (**B**), and 96 (**C**) h. Values are means of five replicates.

**Figure 2 plants-10-01491-f002:**
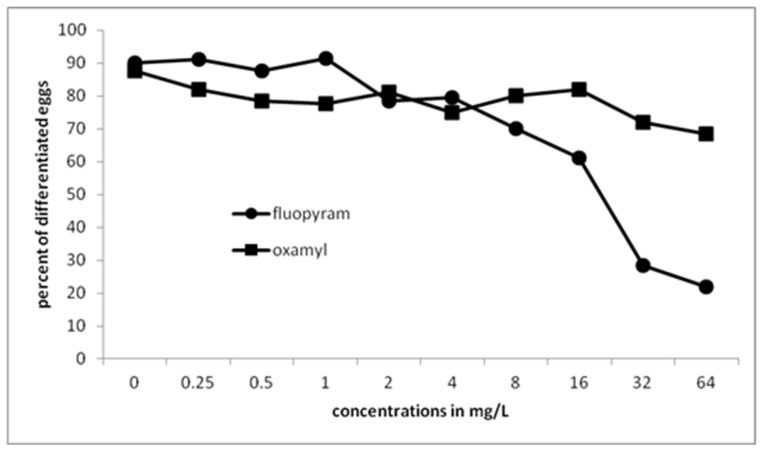
Differentiation of *Meloidogyne javanica* eggs when affected by different doses of fluopyram or oxamyl after immersion of 50 undifferentiated eggs in test solutions for 28 days. Values are means of five replicates.

**Figure 3 plants-10-01491-f003:**
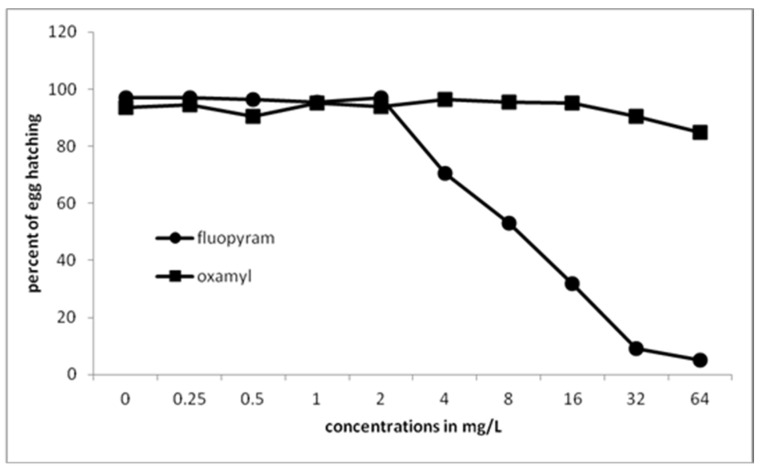
Effect of fluopyram or oxamyl on hatching after immersion of eggs at different concentrations for 30 days. Values are means of five replicates.

**Figure 4 plants-10-01491-f004:**
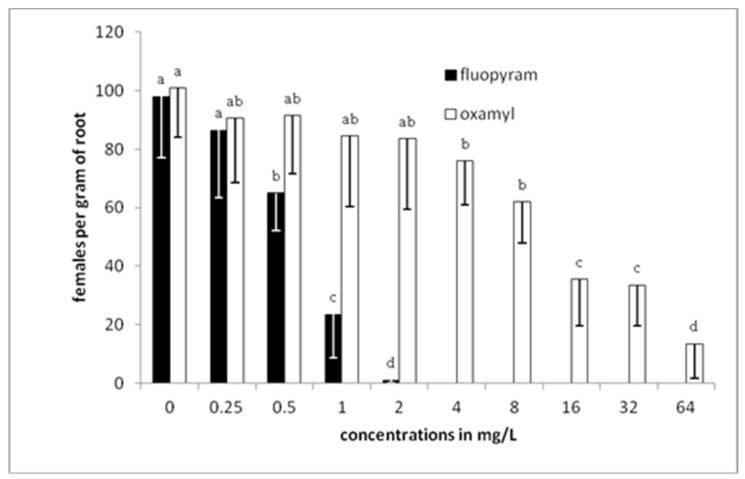
Numbers of females per gram of root after transplanting tomato seedlings in soil treated with either fluopyram or oxamyl and inoculated with 500 J2s. Bars with the same color followed by the same letter are not significantly different at *p* < 0.05. Error bars represent the standard deviation of mean.

**Figure 5 plants-10-01491-f005:**
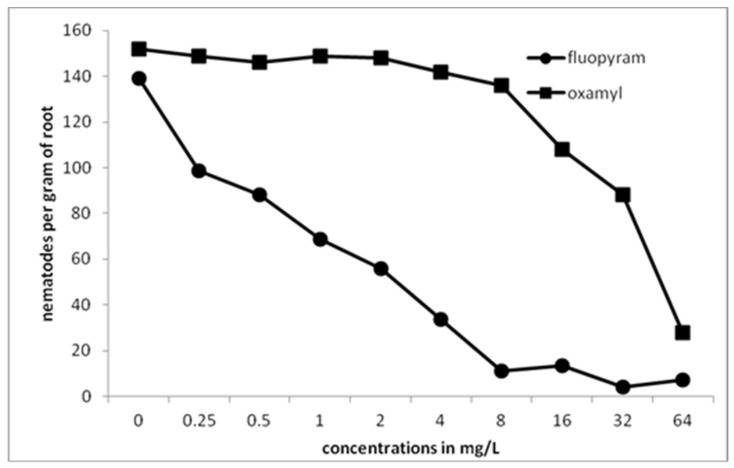
Numbers of all developmental stages of *M. javanica* per gram of root after the treatment of clean tomato roots with different concentrations of fluopyram or oxamyl and transplanted in soil infested with 500 second-stage juveniles.

**Figure 6 plants-10-01491-f006:**
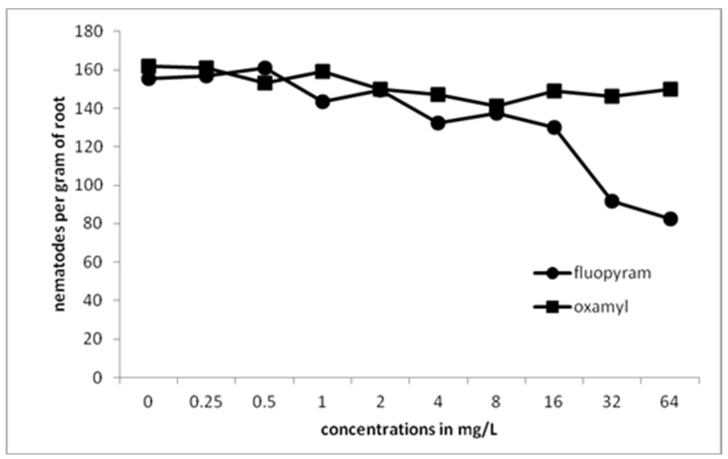
Numbers of all developmental stages of *M. javanica* per gram of root after transplanting tomato plants infected with 500 *Meloidogyne javanica* J2s in soil treated with fluopyram or oxamyl.

**Table 1 plants-10-01491-t001:** Percentage of dead third-stage instars of *Steinernema feltiae* and *Heterhorhabditis bacteriophora* when affected by different concentrations of fluopyram or oxamyl after 24, 48, and 96 h of incubation with 50 third-stage instars per treatment well. Numbers in the same line followed by the same lowercase letters are not significantly different at *p <* 0.05.

	Concentration in mg/L										
		0	0.25	0.5	1	2	4	8	16	32	64
	24 h										
*S. feltiae*	fluopyram	0	0	34c	67b	64b	64b	65b	65b	69b	79a
	oxamyl	1c	1c	2c	4c	2c	5bc	9ab	8ab	10a	12a
*H. bacteriophora*	fluopyram	0	0	0	0	27d	57c	75b	87a	89a	97a
	oxamyl	0	0	0	1b	1b	2b	3ab	3ab	4ab	5a
	48 h										
*S. feltiae*	fluopyram	1e	1e	39d	69bc	67bc	71abc	64c	72abc	74ab	79a
	oxamyl	0f	1f	1f	2f	6e	6e	10dc	13bc	15ab	17a
*H. bacteriophora*	fluopyram	2e	4e	4e	4e	24c	61b	100a	100a	100a	100a
	oxamyl	0	0	0	1b	1b	2ab	3ab	2ab	5ab	5a
	96 h										
*S. feltiae*	fluopyram	2d	32c	81b	85ab	87ab	91a	93a	89ab	87ab	90a
	oxamyl	1f	7e	8de	9de	12cde	13dc	15c	29b	48a	50a
*H. bacteriophora*	fluopyram	1d	71c	75c	86b	100a	100a	100a	100a	100a	100a
	oxamyl	0	0	0	0	3a	3a	3a	5a	5a	7a

## Data Availability

Not applicable.

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
