# Peer review of "Comparison of a Vintage and a Recently Released Nematicide for the Control of Root-Knot Nematodes and Side Effects on Two Entomopathogenic Nematodes"

_plants, 2021, doi:10.3390/plants10081491_

Round 1

Reviewer 1 Report

The manuscript entitled" Comparison of an “old” nematicide with a recently released for the control of Meloidogyne javanica and the site effects on two entomopathogenic nematodes" here authors compared two nematicides, one already in use since long time while the other commercialized few years ago---  the study seems important and justify why new control methods were needed ---- However authors didn’t concluded their study with a section of conclusion which is important part and conveys massage what was achieved and  future implications of the study--- I suggest please include the same---  

  1. Please change the title it looks like to be an newspaper title of some essay but does not reflect like a scientific manuscript title--- please work on it—
  2. Title- what authors wish to convey with site effects- I guess its side effects----
  3. The sentence written by authors that old vs new , talking about newly related nematicide --- I suggest please change wordings to new generation nematicides like (Fluopyram ), the claim that recently released is not sound to be used ----
  4. Introduction- I suggest Although chemical nematicides are providing superior management of nematodes,---
  5. Please avoid using terms old and recently released ---- rephrase them—as authors themselves revealed it was in use since 2017 commercially so recent is not good fit—
  6. Material and methods- Section 4.6.1- Treated tomato plants in infested oil --- what is oil correspond here – I guess it is soil
  7. 7 Section- what was the source for rearing the non-target nematodes- how they were multiplied, please provide details for the same ----

Author Response

The manuscript entitled" Comparison of an “old” nematicide with a recently released for the control of Meloidogyne javanica and the site effects on two entomopathogenic nematodes" here authors compared two nematicides, one already in use since long time while the other commercialized few years ago---  the study seems important and justify why new control methods were needed ---- However authors didn’t concluded their study with a section of conclusion which is important part and conveys massage what was achieved and  future implications of the study--- I suggest please include the same---  (a conclusion was added)

  1. Please change the title it looks like to be an newspaper title of some essay but does not reflect like a scientific manuscript title--- please work on it— (it was changed as asked)
  2. Title- what authors wish to convey with site effects- I guess its side effects---- (site was changed to side)
  3. The sentence written by authors that old vs new , talking about newly related nematicide --- I suggest please change wordings to new generation nematicides like (Fluopyram ), the claim that recently released is not sound to be used ---- (was changed as asked from the two reviewers)
  4. Introduction- I suggest Although chemical nematicides are providing superior management of nematodes,--- (changed as suggested)
  5. Please avoid using terms old and recently released ---- rephrase them—as authors themselves revealed it was in use since 2017 commercially so recent is not good fit— (old was replaced by vintage)
  6. Material and methods- Section 4.6.1- Treated tomato plants in infested oil --- what is oil correspond here – I guess it is soil (oil was changed to soil)
  7. 7 Section- what was the source for rearing the non-target nematodes- how they were multiplied, please provide details for the same ---- (the source of the entomopathogenic nematodes was added)

Reviewer 2 Report

In general, the English needs to be corrected to native standard. As written, there are numerous awkward wordings.

Review of “Comparison of an “old” nematicide with a recently released for the control of Meloidogyne javanica and the site effects on two entomopathogenic nematodes.”

Change title to “Comparison of a vintage and a recently released nematicide for the control of root-knot nematodes and side effects on two entomopathogenic nematodes.”

Here, “old” is unclear in its meaning since it could mean that you have had it in storage, sitting around for a while.  The quotes are not needed. Vintage or classic may be a better word choice.

Abstract:

This first sentence is an example of awkward phrasing of the English. Change to: Root-knot nematodes can cause tremendous losses in vegetable crops. 

Plastic houses versus glasshouses and greenhouse. The preferred term is greenhouse.

The second sentence is also awkward: Farmers usually rely on synthetic nematicides to protect their crops.

Likewise in the third sentence: Recently, newly released nematicides are giving farmers an alternative chemical control for nematodes.

The fourth: In the present study, the efficacy of a vintage nematicide was compared to that of relatively new nematicide, fluopyram.

….and so forth throughout the remainder of the manuscript.

The science in the remaining manuscript of good and the results are new and significant, however, because it is poor written, I reject the manuscript with the recommendation of a major revision, mainly in writing style.

In the revision, please use color in the figures.

Author Response

In general, the English needs to be corrected to native standard. As written, there are numerous awkward wordings.

 Whenever was evident awkward wordings were changed

Review of “Comparison of an “old” nematicide with a recently released for the control of Meloidogyne javanica and the site effects on two entomopathogenic nematodes.”

 Changed as suggested by the reviewer

Change title to “Comparison of a vintage and a recently released nematicide for the control of root-knot nematodes and side effects on two entomopathogenic nematodes.”

 See above

Here, “old” is unclear in its meaning since it could mean that you have had it in storage, sitting around for a while.  The quotes are not needed. Vintage or classic may be a better word choice.

 Old was replaced by vintage

Abstract:

This first sentence is an example of awkward phrasing of the English. Change to: Root-knot nematodes can cause tremendous losses in vegetable crops. 

Plastic houses versus glasshouses and greenhouse. The preferred term is greenhouse.

The second sentence is also awkward: Farmers usually rely on synthetic nematicides to protect their crops.

Likewise in the third sentence: Recently, newly released nematicides are giving farmers an alternative chemical control for nematodes.

The fourth: In the present study, the efficacy of a vintage nematicide was compared to that of relatively new nematicide, fluopyram.

….and so forth throughout the remainder of the manuscript.

 All awkward phrases were rephrased. However the reviewers could provide more suggestions if they think that more phrases could be improved

The science in the remaining manuscript of good and the results are new and significant, however, because it is poor written, I reject the manuscript with the recommendation of a major revision, mainly in writing style.

In the revision, please use color in the figures.

I do not think that colored figures could improve the manuscript. If the reviewer insists then I could make it.

Reviewer 3 Report

Thank you for the interesting manuscript.

The manuscript has merit and is somehow clearly written. All my comments are marked in the attached file. I marked few parts of the manuscript, which needs to be rephrased. Still further English corrections, rephrasing of the text and clarification of the raised points are needed.

Author Response

All corrections and suggestions made by the third reviewer are incorporated in the manuscript.

Round 2

Reviewer 3 Report

I recommend the manuscript for publication.

Author Response

These phrases were added in the materials and methods:
4.3    "Eggs in clean water served as the control."
4.4    "Egg masses submerged in clean water served as the control."
4.5    "Pots under control treatment contained only soil with J2s."
4.6.1    "Pots under control treatment were drenched  with water."
4.6.2    "Pots with soil irrigated with water were served as the control treatment."